# COVID-19 vaccination acceptance among dental students and dental practitioners: A systematic review and meta-analysis

Galvin Sim Siang Lin[1]*, Hern Yue Lee[2◉], Jia Zheng Leong[3◉], Mohammad Majduddin Sulaiman[4‡], Wan Feun Loo[5‡], Wen Wu Tan[6]

1 Department of Dental Materials, Faculty of Dentistry, Asian Institute of Medicine, Science and Technology (AIMST) University, Bedong, Kedah, Malaysia, 2 Seberang Jaya Dental Clinic, Ministry of Health Malaysia, Perai, Pulau Pinang, Malaysia, 3 Petaling Dental Clinic, Ministry of Health Malaysia, Negeri Sembilan, Malaysia, 4 Prosthodontics Unit, School of Dental Sciences, Universiti Sains Malaysia, Health Campus, Kubang Kerian, Kelantan, Malaysia, 5 Bukit Panchor Dental Clinic, Ministry of Health Malaysia, Nibong Tebal, Pulau Pinang, Malaysia, 6 Department of Dental Public Health, Faculty of Dentistry, Asian Institute of Medicine, Science and Technology (AIMST) University, Bedong, Kedah, Malaysia

◉ These authors contributed equally to this work.
‡ MMS and WFL also contributed equally to this work.
* galvin@aimst.edu.my

**Data Availability Statement:** All relevant data are within the paper and its Supporting Information files.

## Abstract

### Background

Dental practitioners and dental students are classified as high-risk exposure to COVID-19 due to the nature of dental treatments, but evidence of their acceptance towards COVID-19 vaccination is still scarce. Hence, this systemic review aims to critically appraise and analyse the acceptability of COVID-19 vaccination among dental students and dental practitioners.

### Materials and methods

This review was registered in the PROSPERO database (CRD42021286108) based on PRISMA guidelines. Cross-sectional articles on the dental students' and dental practitioners' acceptance towards COVID-19 vaccine published between March 2020 to October 2021 were searched in eight online databases. The Joanna Briggs Institute critical appraisal tool was employed to analyse the risk of bias (RoB) of each article, whereas the Oxford Centre for Evidence-Based Medicine recommendation tool was used to evaluate the level of evidence. Data were analysed using the DerSimonian-Laird random effect model based on a single-arm approach.

### Results

Ten studies were included of which three studies focused on dental students and seven studies focused on dental practitioners. Four studies were deemed to exhibit moderate RoB and the remaining showed low RoB. All the studies demonstrated Level 3 evidence. Single-arm meta-analysis revealed that dental practitioners had a high level of vaccination

**Funding:** The authors received no specific funding for this work.

**Competing interests:** The authors have declared that no competing interests exist.

**Abbreviations:** CI, Confidence Interval; COVID-19, Coronavirus Disease 2019; EBSCO, EBSCO Information Services; JBI, Joanna Briggs Institute; LILACS, Latin American and Caribbean Health Science Information Database; NIHR, National Institute for Health Research; OCEBM, Oxford Centre for Evidence-Based Medicine; PRISMA, Preferred Reporting Items for Systematic Reviews and Meta-Analyses Protocols; PROSPERO, Prospective Register of Systematic Reviews; RoB, Risk of Bias; SARS-CoV-2, Severe Acute Respiratory Syndrome Coronavirus 2; WHO, World Health Organisation.

acceptance (81.1%) than dental students (60.5%). A substantial data heterogeneity was observed with the overall $I^2$ ranging from 73.65% and 96.86%. Furthermore, subgroup analysis indicated that dental practitioners from the Middle East and high-income countries showed greater ($p < 0.05$) acceptance levels, while meta-regression showed that the sample size of each study had no bearing on the degree of data heterogeneity.

## Conclusions

Despite the high degree of acceptance of COVID-19 vaccination among dental practitioners, dental students still demonstrated poor acceptance. These findings highlighted that evidence-based planning with effective approaches is warranted to enhance the knowledge and eradicate vaccination hesitancy, particularly among dental students.

## Introduction

A newly identified coronavirus, severe acute respiratory syndrome coronavirus 2 (SARS-CoV-2) or known as the coronavirus disease 2019 (COVID-19), has been wreaking havoc all over the world since its emergence in Wuhan, China in December 2019 [1]. The World Health Organisation (WHO) labelled the COVID-19 outbreak a "Public Health Emergency of International Concern" on January 30, and later, a global pandemic on March 11, 2020 [2]. Ever since, COVID-19 has inflicted millions of deaths globally, presented the government with an unprecedented challenge in the face of severe economic, fiscal, and social pressures, as well as exerted an incredible impact on every sphere of human life.

Despite the implementation of several lockdowns around the world, the infection was not contained due to the reappearance of new COVID-19 variations that were more infective [3]. Several strains of SARS-CoV-2 have been discovered throughout the pandemic and are categorized into three groups: variants of interest, variants of concern, and variants of high consequence [4]. Delta and Omicron variants are among the mutated strains listed as variants of concern which appeared to spread more swiftly than the initial SARS-CoV-2 strain, leading to an increase in COVID-19 cases [4]. These variants have also been linked to increased hospitalisations, reduced neutralisation by antibodies from a previous infection, and diagnostic detection failures [5, 6]. It was soon realised and agreed that herd immunity was the only way to halt the pandemic as several studies have reported promising antibody responses to these variants after vaccine administration [5, 7]. Many countries have begun mass immunisation campaigns for their entire population to curb the widespread of viruses [8]. However, vaccine acceptance and hesitation remained a major obstacle to achieve herd immunity in all countries. Concerns about the vaccination's safety and effectiveness, personal and religious beliefs, and political issues were all mentioned as causes for vaccine apprehension [1, 9].

Dental practitioners are among the healthcare workers classified as high-risk of infection during the COVID-19 pandemic due to the nature of their profession and the close proximity of the dental team to the patients [10]. SARS-CoV-2 spreads rapidly through droplets of saliva during various aerosol-generating dental operations, prompting the development of specific guidelines to minimise virus transmission in the clinical setting [10, 11]. Dental practitioners are also responsible to understand the disease and follow stringent protocols to avoid the spread of disease in their workplaces, assuring the safety of both workers and patients. Nevertheless, vaccination remains the ultimate solution to this issue. On the other hand, dental

students, who make up a small proportion of the oral healthcare workforce, are at the same risk of COVID-19 infection as dental practitioners due to the nature of their clinical training in the dental faculties [12]. It is conceivable that attitudes in the dental profession reflect sentiments in other sectors, leading to a better understanding of vaccine attitudes and the implementation of strategies to tackle vaccine reluctance [13]. Although it has been documented that conspiracy beliefs and misunderstanding about immunity have limited university students' acceptance of the COVID-19 vaccine [14], concrete evidence on COVID-19 acceptability among dental students is still warranted. Thus, determining the vaccination acceptance rate of dental students and dental practitioners is critical.

Vaccine hesitancy is defined as the refusal or postponement of vaccination despite the availability of services [15], and the WHO has identified vaccine hesitancy as a global health threat. Several studies have been undertaken around the world to assess the acceptance rate of the COVID-19 vaccine among dental practitioners and dental students, but the results were ambiguous with a wide array ranging from 56% to 86% [12, 16–20]. To the best of the authors' knowledge, no systematic review has been reported pertaining to the COVID-19 vaccination acceptance among dental students and dental practitioners. Therefore, the present systematic review sought to systematically evaluate the acceptance rate of COVID-19 vaccination among dental students and dental practitioners.

## Materials and methods

### Protocol and registration

The present review followed the Preferred Reporting Items for Systematic Reviews and Meta-Analyses Protocols (PRISMA) guideline [21], and was registered in the Prospective Register of Systematic Reviews (PROSPERO), National Institute for Health Research (NIHR), University of York, with the registration number (ID: CRD42021286108). The focused question was developed by using the PIOT framework, which includes the Population (P), Indicator (I), Outcome of interest (O), and Time (T).

The PIOT criteria were: (1). Population: dental students and dental practitioners (2). Indicator: COVID-19 vaccine (3). Outcome: Acceptance level (4). Time: during COVID-19 pandemic. Hence, the PIOT question was "What is the level of acceptance of COVID-19 vaccine among dental students and dental practitioners?". In this context, a dental student is a person who is currently enrolling in a dental programme and attending a recognised dental school on a regular basis. On the other hand, a dental practitioner is a person who is qualified and licenced by the state law to practise dentistry and provide dental treatments within the limits of their licence and certification. This includes dentists, dental specialists, or postgraduate dental students who have acquired a basic dental degree.

### Search strategy

Three investigators (JZL, HYL, WFL) independently conducted a primary search for articles published between March 2020 and October 2021 using eight electronic databases: Google Scholar, PubMed, Web of Science, Science Direct, Cochrane Library, EBSCO, LILACS, and Open Grey. The reference lists of pertinent articles from the electronic search were independently evaluated by two other investigators (GSSL, MMS) using a computer software (End-Note X9, Thomson Reuters). The following search terms were used for each database: 'acceptance', 'attitude', 'willingness', 'reluctance', 'hesitancy', 'vaccine', 'vaccination', 'dental', 'dentist', 'Covid-19' and 'pandemic'. The Boolean operators 'AND' and 'OR' were used to combine the keywords and construct the search strategy.

## Study selection

After removing duplicate articles using EndNote software version x9, two investigators independently screened the articles based on the title and abstract (JZL, HYL). Following that, another two investigators (WFL, MMS) performed a thorough assessment to identify studies that met the inclusion and exclusion criteria.

The inclusion criteria were: (1). Studies reporting dental students' and dental practitioners' acceptance, reluctance, or hesitancy to receive Covid-19 vaccination; (2). Cross-sectional study; (3). Studies were conducted during the COVID-19 pandemic; (4). No language restriction on published articles. Meanwhile, the exclusion criteria were: (1). Studies that combined data of all healthcare professionals; (2). Case-control, cohort study, expert opinions, reviews, commentaries, editorials, and short communications; (3). Mean and standard deviation on the acceptance or hesitancy level are not reported; (4). Studies conducted before the COVID-19 pandemic. Calibrations between investigators were carried out to assess interrater reliability. The average concordance was calculated with the Kappa value to compare the investigators' decisions on inclusion and exclusion [22]. Any conflicts that arose throughout the search were addressed and resolved with the assistance of the fifth investigator (GSSL).

## Data extraction

The following variables were extracted from each article using a standardised excel spreadsheet form to aid comparability: authors, year of publication, country, type of study, sample size, participant group, gender, age, evaluation tool, response rate and the overall outcomes. One investigator (GSSL) double-checked the accuracy of the data, and any disputes were handled by consensus among all authors.

## Risk of bias assessment

Four investigators (HYL, JZL, WFL, MMS) evaluated the risk of bias for each included study using the Joanna Briggs Institute (JBI) critical appraisal checklist for analytical cross-sectional studies [23]. Either a 'Yes', 'No', 'Unclear' or 'Not Applicable' was assigned for each domain and the studies were categorised as 'Include', 'Exclude' or 'Seek further info'. The Oxford Centre for Evidence-Based Medicine (OCEBM) guideline was employed to determine the level of evidence in each study [24]. The kappa coefficient was used to estimate inter-examiner agreement among all investigators throughout the risk of bias and level of evidence assessments. Besides, any disagreements were handled by discussion among all investigators until a consensus was reached.

## Statistical analysis

All the included primary studies were chosen for quantitative analysis. The weighted mean acceptance rates of COVID-19 vaccination among dental students and dental practitioners from each included study were estimated using a single-arm meta-analysis based on the DerSimonian-Laird random-effects model. The analysis was carried out using the OpenMeta [Analyst] software (CEBM, Oxford, UK) with a significance level of 0.05 and 95% confidence intervals (CI). If the expected upper limit of the 95% confidence interval was larger than 1.0, the upper limit was set to 1.0. The Higgins' $I^2$ statistic was used to identify the degree of data heterogeneity with $I^2$: < 30% = acceptable heterogeneity, $I^2$: 30–60% = moderate heterogeneity, $I^2$: > 60% = substantial heterogeneity [25]. Subgroup analysis and meta-regression were employed to determine the effect of different geographical regions, country income levels and sample sizes on the acceptance rates of COVID-19 vaccination. In addition, Egger's test was employed to investigate publication bias.

# Results

## Study selection

The initial electronic search generated a total of 231 studies. 89 papers were removed after duplication was eliminated, followed by 106 articles that were excluded based on titles and abstracts. The remaining 36 articles were chosen for full-text analysis. Finally, only 10 articles were included in the current review [12, 13, 16–20, 26–28]. The average inter-investigators Kappa score for preliminary article screening (titles and abstracts) and the second screening (full-text assessment) were 0.71 and 0.69, indicating a 'strong' agreement [22]. Fig 1 depicts the reasons for article exclusion, whereas Table 1 summarises the characteristics of the included studies.

All included studies were published in 2021 and employed a cross-sectional design. Four studies were originated from Middle Eastern countries [12, 13, 16, 19], two studies from European countries [17, 20], one study from North America [18], two studies from South Asian countries [27, 28], and one study included participants from 22 different countries [26]. Among them, seven studies explored the acceptance of dental practitioners towards COVID-19 vaccination [13, 16, 17, 19, 20, 27, 28], while the remaining three studies focused on dental students' acceptance or hesitancy [12, 18, 26]. Overall, the response rate ranged from 18% to 81%.

## Risk of bias assessment

Table 2 shows the risk of bias assessment using the JBI critical appraisal tool and the level of evidence for each included study. Generally, four studies were considered moderate risk [12, 16, 17, 28], while the remaining were stated as low risk of bias. All included studies were rated 'Yes' for domains 1, 2, 3, 4, and 5. Three studies were given 'No' for domain 7 [16, 17, 28], whereas four studies were given 'No' for domain 6 [12, 16, 17, 28]. Only one study was deemed 'No' for domain 8 [27]. Furthermore, all studies were rated Level 3 based on the level of evidence due to a lack of blinding among the investigators or assessors. The k coefficients for the risk of bias and level of evidence assessments were 0.68 and 0.75, respectively, indicating a 'strong' agreement.

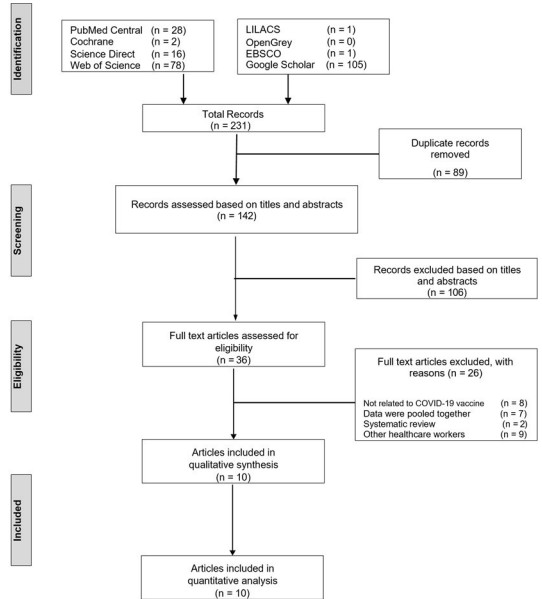

**Fig 1. PRISMA flowchart.** Study selection and reasons for study exclusion according to the PRISMA guidelines.

**Table 1. Characteristics of the included studies.**

| Author | Year | Country | Study design | Sample size | Participant groups | Gender | Age (Mean) | Evaluation tool | Response rate | Results |
|---|---|---|---|---|---|---|---|---|---|---|
| Nasr L *et al.* [16] | 2021 | Lebanon | cross-sectional | 802 | GDP, DSp | Males = 292 Females = 237 | 40.54 ± 14.01 | self-administered questionnaire | 529 (66%) | Already received or willing to receive COVID-19 vaccine (455/529) |
| Belingheri M *et al.* [20] | 2021 | Italy | cross-sectional | 761 | GDP | Male = 301 Female = 120 | N/A | self-administered survey | 421 (55%) | (346/421) 82% declared intent to be vaccinated |
| Mascarenhas AK *et al.* [18] | 2021 | USA | cross-sectional | 1481 | DSt | Male = 42% Female = 58% | 26.8 ± 3.8 | self-administered survey | 238 (16%) | (139/238) 56% willing to receive Vaccine once FDA approved |
| Zigron A *et al.* [13] | 2021 | Israel | cross-sectional | 506 | GDP, PDSt, DSp | Male = 43% Female = 57% | 36.3 | self-administered survey | N/A | overall rate of acceptance for a COVID-19 vaccine: 85% |
| Riad A *et al.* [26] | 2021 | 22 countries | cross-sectional | 6639 | DSt | Male = 1836 Female = 4682 Non-binary = 53 Not disclosed = 68 | 22.06 ± 2.79 | self-administered questionnaire | N/A | Acceptance levels among dental students were found to be 63.5% |
| Papagiannis D *et al.* [17] | 2021 | Greece | cross-sectional | 340 | GDP, MP, P | Male = 51.2% Female—48.8% (*Include all HCW*) | 44.7 ± 10.97 (*Include all HCW*) | self-administered questionnaire | Dentist only—80 (24%) | Dentists reported the highest percentage for Covid-19 vaccine acceptability (82.5%) |
| Kateeb E *et al.* [12] | 2021 | Palestine | cross-sectional | 417 | DSt | Male = 119 Female = 295 Prefer not to say = 3 | N/A | self-administered questionnaire | N/A | 57.8% (n = 241) of the participants are willing to be vaccinated |
| Al-Sanafi M *et al.* [19] | 2021 | Kuwait | cross-sectional | 1019 | GDP, MP, P, Nrs, LT | Male = 101 Female = 69 (*Only GDP*) | 31 ± 7.1 (*Only GDP*) | questionnaire | N/A | 91.2% (155/170) of dentists get or intend to get COVID-19 vaccine |
| Aslam S *et al.* [27] | 2021 | Pakistan | cross-sectional | 370 | GDP | Male = 94 Female = 206 | N/A | self-administered questionnaire | 300 (81%) | 50% (150/300) of the dentists are willing to receive the vaccine |
| Paramashivaih R *et al.* [28] | 2021 | India | cross-sectional | 250 | GDP, DSp, PDSt | Male = 56 Female = 68 | N/A | self-administered questionnaire | 124 (49.6%) | (118/124) of the participants received COVID-19 vaccination |

*GDP: General dental practitioners' DSp: Dental specialists; DSt: Dental students; PDSt: Postgraduate dental students; MP: Medical physicians; P: Pharmacists; Nrs: Nurses; LT: Lab technicians; HCW: Healthcare workers; N/A: Not available.

## Statistical analysis

The acceptances of COVID-19 vaccination among dental students and dental practitioners is presented in Table 3. Meta-analysis was performed when three or more studies are available. Based on the single-arm meta-analysis (Fig 2), the weighted mean acceptance rates of COVID-19 vaccine among dental students and dental practitioners were 60.5% [CI: (56.1, 65.0)] and 81.1% [CI: (72.4, 89.8)], respectively. The $I^2$ of the weighted mean acceptance rates of COVID-19 vaccine among dental students and dental practitioners were 73.65% and 96.86%, respectively, indicating the existence of substantial heterogeneity among the included studies for quantitative analysis.

Sensitivity Analyses were conducted for both dental students and dental practitioners. The highest and lowest weighted mean acceptance rates of COVID-19 vaccine among dental

**Table 2. Risk of bias assessment using the Joanna Briggs Institute (JBI) critical appraisal tool for analytical cross-sectional studies and the level of evidence of each included study.**

| Studies | Domains | | | | | | | | Overall Appraisal | Level of Evidence |
|---|---|---|---|---|---|---|---|---|---|---|
| | 1 | 2 | 3 | 4 | 5 | 6 | 7 | 8 | | |
| Nasr L *et al.* [16] | Y | Y | Y | Y | Y | N | N | Y | Include | 3 |
| Belingheri M *et al.* [20] | Y | Y | Y | Y | Y | Y | Y | Y | Include | 3 |
| Mascarenhas AK *et al.* [18] | Y | Y | Y | Y | Y | Y | Y | Y | Include | 3 |
| Zigron A *et al.* [13] | Y | Y | Y | Y | Y | Y | Y | Y | Include | 3 |
| Riad A *et al.* [26] | Y | Y | Y | Y | Y | Y | Y | Y | Include | 3 |
| Papagiannis D *et al.* [17] | Y | Y | Y | Y | Y | N | N | Y | Include | 3 |
| Kateeb E *et al.* [12] | Y | Y | Y | Y | Y | N | Y | Y | Include | 3 |
| Al-Sanafi M *et al.* [19] | Y | Y | Y | Y | Y | Y | Y | Y | Include | 3 |
| Aslam S *et al.* [27] | Y | Y | Y | Y | Y | Y | Y | N | Include | 3 |
| Paramashivaiah R *et al.* [28] | Y | Y | Y | Y | Y | N | N | Y | Include | 3 |

Domain 1: Were the criteria for inclusion in the sample clearly defined?

Domain 2: Were the study subjects and the setting described in detail?

Domain 3: Was the exposure measured in a valid and reliable way?

Domain 4: Were objective, standard criteria used for measurement of the condition?

Domain 5: Were confounding factors identified?

Domain 6: Were strategies to deal with confounding factors stated?

Domain 7: Were the outcomes measured in a valid and reliable way?

Domain 8: Was appropriate statistical analysis used?.

students were 61.9% [CI: (57.2, 66.6)] and 58.0% [CI: (54.2, 61.8)] when Kateeb E *et al.* [12] and Riad A *et al.* [26] were excluded, respectively. Meanwhile, the highest and lowest weighted mean acceptance rate of COVID-19 vaccine among dental practitioners were 86.3% [CI: (81.5,

**Table 3. Dental students and dental practitioners' acceptance towards COVID-19 vaccination.**

| Studies | Year | Acceptance towards COVID-19 Vaccination | |
|---|---|---|---|
| | | Dental Students | Dental Practitioners |
| Nasr L *et al.* [16] | 2021 | n/a | (455/529) |
| Belingheri M *et al.* [20] | 2021 | n/a | (346/461) |
| Mascarenhas AK *et al.* [18] | 2021 | (139/238) | n/a |
| Zigron A *et al.* [13] | 2021 | n/a | (405/506) |
| Riad A *et al.* [26] | 2021 | (4220/6639) | n/a |
| Papagiannis D *et al.* [17] | 2021 | n/a | (66/80) |
| Kateeb E *et al.* [12] | 2021 | (241/417) | n/a |
| Al-Sanafi M *et al.* [19] | 2021 | n/a | (155/170) |
| Aslam S *et al.* [27] | 2021 | n/a | (150/300) |
| Paramashivaiah R *et al.* [28] | 2021 | n/a | (118/124) |

*n/a: Not Available.

Nasr L *et al.* pooled data for general dentists and dental specialists.

Zigron A *et al.* pooled data for general dentists and dental specialists.

Riad A *et al.* students that answered, 'totally agree' and 'agree' will be deemed as acceptance towards COVID-19 vaccination.

Kateeb E *et al.* pooled data for dental students and dental fresh graduates.

Paramashivaiah R *et al.* pooled data for general dentists, dental specialists and postgraduate dental students.

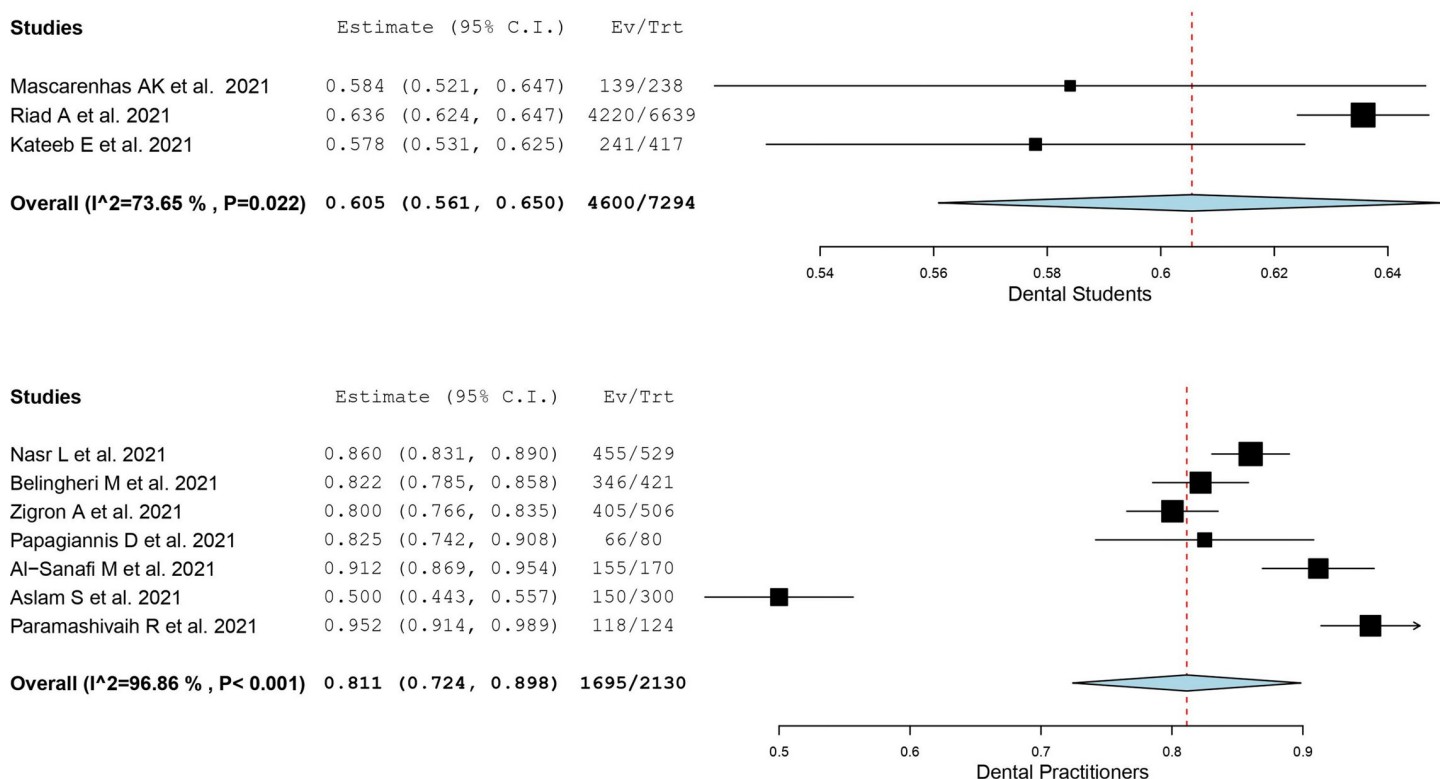

**Fig 2. Meta-analysis of COVID-19 vaccine acceptance.** Single-arm meta-analyses showing the weighted mean acceptance rates of COVID-19 vaccination among dental students and dental practitioners.

91.1)] and 78.8% [CI: (69.5, 88.0)] when Aslam S *et al.* [27] and Paramashivaih R *et al.* [28] were omitted, respectively.

Considering the sheer degree of data heterogeneity, subgroup analyses were performed to determine the impact of different geographical regions and country income levels on COVID-19 vaccine acceptance rates (S1 Table). Data were classified into four geographical regions: Middle East, Europe, North America, and South Asia (depending on where each included study was conducted), and they were divided into three categories: high, upper-middle, and lower-middle based on their respective country income levels. Since study done by Riad A *et al.* [26] was eliminated due to the extensive pooling of respondents from multiple countries, subgroup analysis on dental students' acceptance level was not undertaken as only two remaining studies were left [12, 18]. On the other hand, dental practitioners in the Middle East had a considerably higher weighted mean acceptance rate of COVID-19 vaccination ($p < 0.001$) than those in Europe and South Asia. Meanwhile, dental practitioners from high-income countries demonstrated significantly higher acceptability of the COVID-19 vaccine ($p < 0.001$) than those from upper-middle and lower-middle-income countries. Nonetheless, subgroup analyses on the effect of gender and participants' age on the acceptance level were not possible in the current review since the data were aggregated in the primary studies. It is also not feasible to divide dental practitioners into subgroups such as general dentists, dental specialists, or postgraduate dental students due to a paucity of data.

Meta-regression was performed to evaluate the effect of the response sample sizes of each study on the acceptance rate of the COVID-19 vaccine (S2 Table). No significant differences were found for both dental students ($p = 0.06$) and dental practitioners ($p = 0.611$), signifying that the sample size of each study does not have any direct effect on the degree of data

heterogeneity. Egger's test revealed that there was no evidence of significant publication bias in the acceptance rates of COVID-19 vaccine among dental students and dental practitioners. (Egger's test: p-value = 0.11, and 0.08, respectively).

## Discussion

The present systematic review aimed to comprehensively investigate the acceptance of COVID-19 vaccination among dental students and dental practitioners in order to provide valuable insights for future COVID-19 vaccination implementation. Based on the current single-arm meta-analysis, dental practitioners showed a high acceptance rate towards COVID-19 vaccination (81.1%) which is higher than the values reported in previous systematic reviews conducted on healthcare workers that ranged from 51% to 73% [8, 29]. The authors speculated that the disparities in acceptance rates could be explained by the passage of time [19], as previous reviews included studies or surveys conducted at a period when comprehensive scientific evidence on COVID-19 was not yet available and healthcare professionals' acceptance of the vaccine at that moment was still confined. Conversely, the primary studies included in the present review were recently published in the year 2021. Such differences can also be explained by the increased awareness of the high infectivity of COVID-19 disease and its ability to induce more severe illnesses and complications [30]. Also, COVID-19 vaccine mandates among healthcare workers implemented by various countries in recent months may have increased the acceptance rate of COVID-19 vaccine among healthcare workers.

Acknowledging dental practitioners' perceptions on COVID-19 vaccination is critical as they play a key role in educating patients and tackling vaccine reluctance among the general public [19]. Undeniably, dental practitioners were exposed to a high risk of cross-infection in clinical practice, and preventive measures such as proper use of personal protective equipment when performing aerosol-generating procedures have raised awareness of the importance of mitigating the spread of infection, resulting in increased vaccination acceptance [16, 20]. Despite the significant likelihood of being infected by the viruses, the prevalence of COVID-19 infection among dental practitioners was reported to be low with a prevalence rate of 2.6% in the United States and 1.9% in France, respectively [31, 32]. The authors speculated that this could be attributed to the improved preventive measures, as well as the increased vaccine acceptability and uptake.

Moreover, most countries also mandated vaccination policies for healthcare professionals which may have an impact on dental practitioners' acceptability towards COVID-19 vaccination [33]. Other factors such as the knowledge and concern of being infected with the SARS-CoV-2 may render dental practitioners to accept vaccination [16, 27]. In fact, several studies have proven that prior influenza vaccination history has a significant influence on the acceptability of covid-19 vaccination [34, 35], and this could be the driving force for the high degree of acceptance among dental practitioners. Nonetheless, it should be highlighted that vaccination acceptance may have been overestimated as dental practitioners who were not interested in receiving the vaccine may be unlikely to participate in the survey or questionnaire [16].

Subgroup analysis suggested that both geographical regions and country income levels significantly affect the acceptance of vaccination among dental practitioners. It was discovered that dental practitioners in the Middle East had a considerably higher acceptance rate of COVID-19 vaccination compared to other regions. This could be attributed to the surge of COVID-19 cases and the increase in mortality rate (especially among dentists and physicians) in the Middle East region at the time of the surveys [16, 19]. The authors postulated that dental practitioners' perspectives about the coronavirus have shifted from reluctance to acceptance as a result of their fear of infection [20]. On the other hand, a low acceptance rate of the COVID-19 vaccination in South Asia, particularly in Pakistan, was primarily owing to public figures'

conspiracy theories, lack of confidence in locally manufactured vaccines and lack of effort by public health authorities to educate dental healthcare workers [27]. Thus, tackling the underlying cultural and political factors that cause vaccine hesitancy is crucial to boost vaccination uptake. The present review also showed that the country's income level was a significant determinant of the acceptance rate of COVID-19 vaccination among dental practitioners. This is consistent with a previous systematic review reporting that the highest acceptance rate of COVID-19 vaccine among healthcare workers was from high-income countries, while the lowest acceptance rate was seen from low-income countries [14]. Furthermore, despite growing evidence of the safety and effectiveness of presently used vaccines, one explanation for the disparity in COVID-19 vaccine uptake across countries might be attributed to the vaccine's availability [36]. Since vaccine development takes time, administrating approved vaccines to a wide population would be challenging. Nonetheless, it can be predicted that vaccination uptake will continue to rise as vaccine availability increases [37].

In contrast to dental practitioners, dental students in the present analysis had a lower weighted mean acceptance rate (60.5%) towards COVID-19 vaccination. Their willingness and readiness to accept the vaccine were found to be greatly reduced due to a lack of trust in the government and vaccination data from the pharmaceutical sectors [12, 26]. Most students also did not consider themselves at risk of being infected by the coronavirus, as the available evidence reported that the COVID-19 infection rate among dentists in the United States dentist was as low as 0.9% [18, 38]. Other contributing factors to the poor acceptance rate among dental students included socioeconomic status, perceived COVID-19 vaccination knowledge, and gender [16, 18]. Nonetheless, subgroup analysis was not performed among dental students due to a paucity of published articles. It is still worth noting that understanding the underlying causes of vaccination apprehension, especially among dental students, can pave the way for higher education institutions to reinforce immunization knowledge and understanding.

The current review used the JBI critical appraisal checklist for cross-sectional studies to critically evaluate the risk of bias in each included study. The JBI risk of bias assessment tool is among the most widely used tools for assessing the internal validity of the included studies due to its ease of comprehension and implementation [39]. Four studies were rated 'No' for domain 6 (Were strategies to deal with confounding factors stated?) [12, 16, 17, 28]. Confounding factors such as the respondents' gender and age should be addressed to prevent the true effects of the study from being concealed. As there may be disparities in the confounding factors taken into consideration, determining what the researchers performed to adjust the confounders could be challenging. Restriction and stratification are two approaches to deal with confounders. Researchers may, for instance, limit their study to a specific age group of respondents or divide the population into strata or subgroups for comparison [40]. Overlooked confounders might increase the odds for findings to be skewed and their validity to be questioned. Thus, confounding factors must be carefully identified and dealt with to optimize reliability.

In addition, three studies failed to measure the outcomes in a valid and reliable way as they did not mention whether the questionnaires or surveys used were pre-validated [16, 17, 28]. For a questionnaire or survey to be considered acceptable, it must have two key characteristics: reliability and validity. It is advisable that qualified experts should be involved in the face validation process to evaluate each questionnaire item, followed by a pilot test on a small sample of respondents. Moreover, the authors recommended that content and construct validity should be explored when validating a questionnaire [41]. Only one study failed to specify appropriate statistical analysis used [27]. The choice of an appropriate statistical approach is critical, as it will influence the overall outcome. The present review also reported no evidence of significant publication bias among the articles included, which could be owing to the extensive literature search that included grey literature.

It can be concluded that dental practitioners, particularly those from the Middle East regions and high-income countries, demonstrated high acceptability of COVID-19 vaccination. While dental students showed a lower acceptance towards COVID-19 vaccination, future research should concentrate on the reported unfavourable attitudes and hesitation factors. Furthermore, scientific research on the impact of geographical regions and country economic levels on vaccination acceptability levels among dental students should be explored. It is imperative to eradicate vaccination hesitancy among dental students and dental practitioners, given their critical role in educating the public on the awareness and importance of vaccination. Establishing immunisation campaign strategies and implementing courses or modules on vaccination literacy in the dental curriculum should be prioritised in the fight against the pandemic. Meanwhile, it is also essential to keep records of how dental students and dental practitioners respond to vaccinations and adjust vaccination strategies as required.

## Strengths and limitations

The strengths of the current systematic review include the registration of the study protocol in the PROSPERO database for better transparency and preclude incidental review duplication [42], a comprehensive literature search was performed in eight electronic databases to ensure that no relevant articles were missed, the literature search and data extraction were carried out by several independent investigators, and a well-designed risk of bias assessment tool was used to appraise the included articles [43]. Moreover, another merit of the present review is that there are no language constraints on the included articles. Merely including specific language papers will limit the breadth of the studies included and introduce significant language bias into the current review [44]. To account for between-study variability, a random-effects model based on the DerSimonian-Laird was adopted in the current analysis as the authors considered that the true effect size varied among studies since each study is different. Furthermore, a random-effects model can also help to account for the significant heterogeneity identified in the present review [45]. Several drawbacks were identified in the current study. First, the limited amount of studies included in the present meta-analysis may hinder the investigators from obtaining reliable inferential outcomes [46]. However, it is understandable that such a prerequisite of including a large number of studies is rarely achieved, notably in the field of dentistry [47]. Another flaw in the current systematic review is the degree of precision in data synthesis, as the analysis was not feasible for respondents' age, gender, or past vaccination history, all of which could influence the overall results. Although subgroup analyses were conducted to assess the effect of geographical regions and country income levels on dental practitioners' COVID-19 vaccine acceptance rates, no analysis concerning dental students was carried out as the research included were limited. Also, different studies utilised different questionnaires or surveys, which could have resulted in response bias due to the lack of general standardisation. Another shortcoming is that the majority of the studies are from the Middle East, Europe, and South Asia. Therefore, drawing solid conclusions on the global acceptance of the COVID-19 vaccine is impractical, and more well-designed cross-sectional studies focusing on the perceptions of oral health professionals from other regions are warranted.

## Supporting information

**S1 Table. Subgroup analyses of geographical regions and country income levels on the acceptance rates (%) of COVID-19 vaccine among dental practitioners.**
(PDF)

**S2 Table. Meta-regression evaluating the effect of sample size of each study on the acceptance rates (%) of COVID-19 vaccine among dental students and dental practitioners.**
(DOCX)

**S1 Checklist. PRISMA 2020 for abstracts checklist.**
(DOCX)

**S2 Checklist. PRISMA checklist.**
(DOCX)

## Author Contributions

**Conceptualization:** Galvin Sim Siang Lin.

**Data curation:** Galvin Sim Siang Lin, Hern Yue Lee, Jia Zheng Leong, Mohammad Majduddin Sulaiman, Wan Feun Loo.

**Formal analysis:** Galvin Sim Siang Lin.

**Methodology:** Galvin Sim Siang Lin.

**Software:** Galvin Sim Siang Lin.

**Visualization:** Wen Wu Tan.

**Writing – original draft:** Hern Yue Lee, Jia Zheng Leong, Mohammad Majduddin Sulaiman, Wan Feun Loo.

**Writing – review & editing:** Galvin Sim Siang Lin, Wen Wu Tan.

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
