## [Decision Letter · Decision Letter 0]

8 Feb 2022

PONE-D-22-00027COVID-19 Vaccination Acceptance Among Dental Students and Dental Practitioners: A Systematic Review and Meta-AnalysisPLOS ONE

Dear Dr. Lin,

Thank you for submitting your manuscript to PLOS ONE. After careful consideration, we feel that it has merit but does not fully meet PLOS ONE’s publication criteria as it currently stands. Therefore, we invite you to submit a revised version of the manuscript that addresses the points raised during the review process.

We look forward to receiving your revised manuscript.

Kind regards,

Sanjay Kumar Singh Patel, Ph.D.

Academic Editor

PLOS ONE

Journal Requirements:

Reviewers' comments:

Reviewer's Responses to Questions

**Comments to the Author**

1. Is the manuscript technically sound, and do the data support the conclusions?

Reviewer #1: Yes

Reviewer #2: Yes

2. Has the statistical analysis been performed appropriately and rigorously? 

Reviewer #1: Yes

Reviewer #2: Yes

3. Have the authors made all data underlying the findings in their manuscript fully available?

Reviewer #1: Yes

Reviewer #2: No

4. Is the manuscript presented in an intelligible fashion and written in standard English?

Reviewer #1: Yes

Reviewer #2: Yes

5. Review Comments to the Author

Reviewer #1: In this paper entitled "COVID-19 Vaccination Acceptance Among Dental Students and Dental Practitioners: A Systematic Review and Meta-Analysis", the authors investigated the acceptance of COVID-19 vaccination among dental students and dental practitioners. The authors took cross-sectional articles on dental students and dental practitioners in the manuscript and then performed the statistical analysis. The manuscript is easy to understand and has been completed with statistical rigor. But there are few problems with the manuscript.

Minor Comments:

1) Add more information in the background section, such as the purpose of study, etc.

2) Introduction: Minor information on the variants of COVID-19 and their future challenges can be included i.e. doi: 10.1007/s15010-021-01734-2.

3) The English may be improved (Minor).

4) The graphical figure resolution is low. It has to be improved for publication. It isn't easy to interpret results from them.

5) Proper figure legends are also missing from the manuscript.

6) The authors should cross-check all abbreviations in the manuscript. Provide all abbreviations in a separate paragraph.

7)Add the limitation of the manuscript in a separate section.

Reviewer #2: The manuscript is well written, and it can be accepted after the minor revision. Please find my comments below.

1. Introduction, please include some quantitative information about COVID-19 variants and their challenges in its prevention.

2. Discussion, please highlight minor information about cases, mortality, and casualty of COVID-19 in case of dental students and dental practitioners.

3. Please include 2 or 3 figures at least to represent your results in the revised manuscript.

---

## [Author Response · Author response to Decision Letter 0]

16 Feb 2022

Reviewer 1

1. Add more information in the background section, such as the purpose of study, etc. 

Reply: 

The background has been amended.

Page 2, Line 3-6:

‘The contemporary COVID-19 pandemic has prompted researchers across the world …… acceptance towards COVID-19 vaccination is still scarce.’

2. Introduction: Minor information on the variants of COVID-19 and their future challenges can be included i.e., Doi: 10.1007/s15010-021-01734-2. 

Reply:

Several points have been added to the introduction.

Page 4:

“Several strains of SARS-CoV-2 have been discovered throughout the pandemic and are categorized into three groups …… to curb the widespread of the viruses”

3. The English may be improved (Minor). 

Reply: The manuscript has been proofread by an English native speaker.

4. The graphical figure resolution is low. It has to be improved for publication. It isn't easy to interpret results from them. 

Reply: The authors would like to thank the reviewer for raising this concern. High-resolution figures (Figure 1 and Figure 2) will be provided to the journal editor.

5. Proper figure legends are also missing from the manuscript. 

Reply: Figure legends were written in the text. 

Page 9:

“Fig 1. PRISMA flowchart. Study selection …… PRISMA guidelines”

Page 13

“Fig 2. Meta-analysis of COVID-19 vaccine acceptance …… and dental practitioners.”

6. The authors should cross-check all abbreviations in the manuscript. Provide all abbreviations in a separate paragraph.

Reply: 

All abbreviations have been cross-checked and a separated section ‘Abbreviation’ was added into the manuscript.

7. Add the limitation of the manuscript in a separate section. 

Reply:

A separated section of limitation has been added.

Reviewer 2

1. Introduction, please include some quantitative information about COVID-19 variants and their challenges in its prevention. 

Reply: 

Several points have been added to the introduction.

Page 4:

“Several strains of SARS-CoV-2 have been discovered throughout the pandemic and are categorized into three groups …… to curb the widespread of the viruses”

2. Discussion, please highlight minor information about cases, mortality, and casualty of COVID-19 in case of dental students and dental practitioners. 

Reply:

The authors added few points in the discussion.

Page 16:

“Despite the significant likelihood of being infected by the viruses, …… the increased vaccine acceptability and uptake.”

3. Please include 2 or 3 figures at least to represent your results in the revised manuscript. 

Reply:

Two figures were included in the manuscript:

Figure 1 – PRISMA flowchart showing the study selection

Figure 2 – Single-arm meta-analysis showing the outcome of the present review

---

## [Decision Letter · Decision Letter 1]

10 Mar 2022

PONE-D-22-00027R1COVID-19 Vaccination Acceptance Among Dental Students and Dental Practitioners: A Systematic Review and Meta-AnalysisPLOS ONE

Dear Dr. Lin,

Thank you for submitting your manuscript to PLOS ONE. After careful consideration, we feel that it has merit but does not fully meet PLOS ONE’s publication criteria as it currently stands. Therefore, we invite you to submit a revised version of the manuscript that addresses the points raised during the review process.

Having intensively reviewed your revised draft, our external reviewers basically have agreed with their final recommendations. Additionally, I have double checked your re-submitted version, to come to a more balanced decision (see R #3). All in all, I am convinced that your re-revised paper will be worth following, even if your revised version still would benefit from thorough re-edits and language polishing. Thus I would like to encourage you to provide a thorough (in terms of language, reviewers' constructive criticism, content, generalizable outcome, and/or Authors' Guidelines) revision in order to avoid an iterative and lengthy review process and facilitate a smooth publication process.

We look forward to receiving your revised manuscript.

Kind regards,

Andrej M Kielbassa, Prof. Dr. med. dent. Dr. h. c.

Academic Editor

PLOS ONE

Journal Requirements:

Reviewers' comments:

Reviewer's Responses to Questions

**Comments to the Author**

1. If the authors have adequately addressed your comments raised in a previous round of review and you feel that this manuscript is now acceptable for publication, you may indicate that here to bypass the “Comments to the Author” section, enter your conflict of interest statement in the “Confidential to Editor” section, and submit your "Accept" recommendation.

Reviewer #1: All comments have been addressed

Reviewer #2: All comments have been addressed

Reviewer #3: (No Response)

2. Is the manuscript technically sound, and do the data support the conclusions?

Reviewer #1: Yes

Reviewer #2: Yes

Reviewer #3: No

3. Has the statistical analysis been performed appropriately and rigorously? 

Reviewer #1: Yes

Reviewer #2: Yes

Reviewer #3: Yes

4. Have the authors made all data underlying the findings in their manuscript fully available?

Reviewer #1: Yes

Reviewer #2: Yes

Reviewer #3: Yes

5. Is the manuscript presented in an intelligible fashion and written in standard English?

Reviewer #1: Yes

Reviewer #2: Yes

Reviewer #3: Yes

6. Review Comments to the Author

Reviewer #1: In this manuscript entitled "COVID-19 Vaccination Acceptance Among Dental Students and Dental Practitioners: A Systematic Review and Meta-Analysis " the authors have addressed all the previous concerns. Therefore, the manuscript has no deficiency so it can be published in the Journal.

Reviewer #2: (No Response)

Reviewer #3: Abstract

- With 334 words, this section is much too long. Please stick to the Journal guidelines, and reduce to the word maximum which is 300.

- With your revision, give priority to the results.

- With your Conclusions, please stick exclusively to your revised aims. Do not simply repeat your results here. Instead, provide a reasonable and generalizable extension of your outcome.

Intro

- Would seem sound.

- Please add some thoughts on your rationale. Dentists/dental students are considered a part of the society. Consequently, why should they differ from the society? Please clarify your idea when starting this project.

Meths

- Revision would seem satisfying.

Results

- "Finally, only 10 articles are included (...)" must read "Finally, only 10 articles were included (...)".

- Level of evidence of included studies would seem poor. Please comment in this aspect. Why did you include such poor papers? And why have those poor papers been published? Please discuss thoroughly.

- Again, please stick to Journal guidelines. "(P=0.006)" must read "(p = 0.006)". "p" must be italicized.

- "No significant differences were found for both dental students (P=0.006) (...)." "(p = 0.006)" would indicate a significant difference, don't you think so?

Disc

- "To the best of the authors’ knowledge, it is also the first of its kind to critically summarise, analyse, and provide reliable evidence-based findings from existing research on the acceptability of COVID-19 vaccination in the dental community." Don't be too proud of your work, and do not re-repeat your pride. Compare to your Intro section ("To the best of the authors’ knowledge, no systematic review has been reported (...)"), and revise carefully. All readers will acknowledge your work, even if you will not show "the best of your knowledge"...

- "(...) dental practitioners showed a high acceptance rate towards COVID-19 vaccination (81.1%) which is higher than the values reported in previous systematic reviews conducted on healthcare workers that ranged from 51% to 73% [8, 29]." I do not agree that this would be a "high acceptance rate". First, dental practitioners are a part of the medical science community, and this would call for a considerably higher rate. Second, there are countries with much higher rates, even with the normal population (see, for example, Arabic Emirates, Chile, or Portugal). Consequently, there is no need to be satisfied. Please discuss.

- "(...) a comprehensive literature search was performed in eight electronic databases to ensure that no relevant articles were missed." Please note that there are more databases. What about other languages? Please provide a sound assessment of the probability on how many papers have been missed, and how this could influence your outcome.

- "Therefore, drawing solid conclusions on the global acceptance of the COVID-19 vaccine is impractical (...)." First, this would mean that publishing your paper will not be advocated. Second, why did you include a Conclusion section ("it can be concluded that dental practitioners, (...).") here?

Concl

- Again, see comments given above, and revise carefully.

- Phrases given with "Furthermore, scientific research on the impact of (...)" to "(...) it is also essential to keep records of how dental students and dental practitioners respond to vaccinations and adjust vaccination strategies as required." would seem right, but are not considered Conclusions and must be copied & pasted to the Disc section.

Figs

- Prisma Flow Chart not visible with your PDF. Please recreate this Figure by using PowerPoint, for example.

In total, this revised and re-submitted draft is not considered ready to proceed, and further revisions would see mandatory.

7. PLOS authors have the option to publish the peer review history of their article (what does this mean?). If published, this will include your full peer review and any attached files.

Reviewer #1: **Yes: **Aditya Kumar Sharma

Reviewer #2: No

Reviewer #3: No

---

## [Author Response · Author response to Decision Letter 1]

11 Mar 2022

1. Please review your reference list to ensure that it is complete and correct. If you have cited papers that have been retracted, please include the rationale for doing so in the manuscript text or remove these references and replace them with relevant current references. Any changes to the reference list should be mentioned in the rebuttal letter that accompanies your revised manuscript. If you need to cite a retracted article, indicate the article’s retracted status in the References list and also include a citation and full reference for the retraction notice.

Reply: The reference list was amended accordingly. 

Reviewer 3

1. Abstract:

With 334 words, this section is much too long. Please stick to the Journal guidelines and reduce to the word maximum which is 300.

- With your revision, give priority to the results.

- With your Conclusions, please stick exclusively to your revised aims. Do not simply repeat your results here. Instead, provide a reasonable and generalizable extension of your outcome.

Reply: Dear reviewer, the authors attempted to reduce the number of words, and we agreed to use the full name, such as Joanna Briggs Institute, rather than JBI. Hence, the word count may slightly exceed.

The authors did not repeat the results, but we provide a generalized outcome:

“Despite the high degree of acceptance of COVID-19 vaccination among dental practitioners, dental students still demonstrated a poor acceptance”

An extension of the outcome was also provided:

“These findings highlighted that evidence-based planning with effective approaches is warranted to enhance the knowledge and eradicate vaccination hesitancy, particularly among dental students”

2. Intro

- Would seem sound.

- Please add some thoughts on your rationale. Dentists/dental students are considered a part of the society. Consequently, why should they differ from the society? Please clarify your idea when starting this project.

Reply: The authors have addressed the reason for conducting such a study because dental practitioners and dental students are classified as high-risk group.

“Dental practitioners are among the healthcare workers classified as high-risk of infection during the COVID-19 pandemic due to the nature of their profession and the close proximity of the dental team to the patients”

Therefore, they should be distinguished from the general public or society in order to determine the real impact of COVID-19 vaccination acceptance among them.

3. Results

- "Finally, only 10 articles are included (...)" must read "Finally, only 10 articles were included (...)".

- Level of evidence of included studies would seem poor. Please comment in this aspect. Why did you include such poor papers? And why have those poor papers been published? Please discuss thoroughly.

- Again, please stick to Journal guidelines. "(P=0.006)" must read "(p = 0.006)". "p" must be italicized.

- "No significant differences were found for both dental students (P=0.006) (...)." "(p = 0.006)" would indicate a significant difference, don't you think so?

Reply: The word ‘are’ had been changed to ‘were’.

The authors would like to thank the reviewer for raising this concern. Due to the nature of the study design, the level of evidence of the included cross-sectional studies appeared to be poor, as none of the research involved blinding the investigators or assessors.

Nonetheless, the included cross-sectional studies are still valuable as they can aid in determining the degree of acceptance of COVID-19 vaccination during this specific period.

The author also added the phrase:

“…due to a lack of blinding among the investigators or assessors”

Please accept the authors' sincere apologies for the 'p = 0.06' error.

5. - "To the best of the authors’ knowledge, it is also the first of its kind to critically summarise, analyse, and provide reliable evidence-based findings from existing research on the acceptability of COVID-19 vaccination in the dental community." Don't be too proud of your work, and do not re-repeat your pride. Compare to your Intro section ("To the best of the authors’ knowledge, no systematic review has been reported (...)"), and revise carefully. All readers will acknowledge your work, even if you will not show "the best of your knowledge"... 

Reply: The authors have deleted the sentence in the first paragraph of the discussion section.

6. "(...) dental practitioners showed a high acceptance rate towards COVID-19 vaccination (81.1%) which is higher than the values reported in previous systematic reviews conducted on healthcare workers that ranged from 51% to 73% [8, 29]." I do not agree that this would be a "high acceptance rate". First, dental practitioners are a part of the medical science community, and this would call for a considerably higher rate. Second, there are countries with much higher rates, even with the normal population (see, for example, Arabic Emirates, Chile, or Portugal). Consequently, there is no need to be satisfied. Please discuss. 

Reply: The authors would like to express their gratitude to the reviewer for sharing their thoughts on this matter. However, the authors believe that different religions and beliefs about COVID-19 immunization do exist, as detailed in the introduction section. It is also unfair to assume that all dental practitioners across the world, although being part of the medical community, would have a high acceptance rate.

The authors were not particularly proud with the 81.8% acceptance rate; rather, we were comparing the acceptance rate to data reported among healthcare personnel in order to provide a more comprehensive picture to the readers.

7. "(...) a comprehensive literature search was performed in eight electronic databases to ensure that no relevant articles were missed." Please note that there are more databases. What about other languages? Please provide a sound assessment of the probability on how many papers have been missed, and how this could influence your outcome.

Reply: The authors did not impose any language restriction and the limitations were addressed in the manuscript.

8. Therefore, drawing solid conclusions on the global acceptance of the COVID-19 vaccine is impractical (...)." First, this would mean that publishing your paper will not be advocated. Second, why did you include a Conclusion section ("it can be concluded that dental practitioners, (...).") here? 

Reply: The authors intended to provide recommendations for future research, advocating that more comparable studies be undertaken in order to reach a solid conclusion.

Of course, this does not imply that the existing manuscript has no value or merit. The authors believe that publishing the current data, it would raise awareness among dental researchers throughout the globe to delve deeper into this context.

Furthermore, conducting a systematic review allows readers to recognize the existing limitations.

The conclusion section was eliminated, and the points were relocated to the last paragraph of the discussion section.

9 Again, see comments given above, and revise carefully.

- Phrases given with "Furthermore, scientific research on the impact of (...)" to "(...) it is also essential to keep records of how dental students and dental practitioners respond to vaccinations and adjust vaccination strategies as required." would seem right, but are not considered Conclusions and must be copied & pasted to the Disc section. 

Reply: The conclusion section was eliminated, and the points were relocated to the last paragraph of the discussion section.

---

## [Decision Letter · Decision Letter 2]

1 Apr 2022

PONE-D-22-00027R2COVID-19 Vaccination Acceptance Among Dental Students and Dental Practitioners: A Systematic Review and Meta-Analysis

PLOS ONE

Dear Dr. Lin,

Thank you for re-submitting your revised manuscript to PLOS ONE. After careful consideration, we feel that it has merit but does not fully meet PLOS ONE’s publication criteria as it currently stands. Therefore, we invite you to submit a revised version of the manuscript that addresses the points raised during the review process.

Having intensively reviewed your revised draft, our external reviewers differed to some extent with their final recommendations. Additionally, I have double checked your re-submitted version, to come to a more balanced decision (see R #1). All in all, I am convinced that your re-revised paper will be worth following, even if your current version still would benefit from thorough re-edits and some language polishing. Thus, I would like to encourage you to provide a thorough (in terms of language, reviewers' constructive criticism, content, generalizable outcome, and/or Authors' Guidelines) revision in order to avoid an iterative and lengthy review process and facilitate a smooth publication process.

We look forward to receiving your revised manuscript.

Kind regards,

Andrej M Kielbassa, Prof. Dr. med. dent. Dr. h. c.

Academic EditorPlos One  Journal Requirements:

Reviewers' comments:

Reviewer's Responses to Questions

**Comments to the Author**

1. If the authors have adequately addressed your comments raised in a previous round of review and you feel that this manuscript is now acceptable for publication, you may indicate that here to bypass the “Comments to the Author” section, enter your conflict of interest statement in the “Confidential to Editor” section, and submit your "Accept" recommendation.

Reviewer #1: (No Response)

Reviewer #2: All comments have been addressed

Reviewer #3: All comments have been addressed

2. Is the manuscript technically sound, and do the data support the conclusions?

Reviewer #1: Yes

Reviewer #2: Yes

Reviewer #3: Yes

3. Has the statistical analysis been performed appropriately and rigorously? 

Reviewer #1: Yes

Reviewer #2: N/A

Reviewer #3: Yes

4. Have the authors made all data underlying the findings in their manuscript fully available?

Reviewer #1: Yes

Reviewer #2: Yes

Reviewer #3: Yes

5. Is the manuscript presented in an intelligible fashion and written in standard English?

Reviewer #1: Yes

Reviewer #2: Yes

Reviewer #3: Yes

6. Review Comments to the Author

Reviewer #1: In this manuscript, "COVID-19 Vaccination Acceptance Among Dental Students and Dental Practitioners: A Systematic Review and Meta-Analysis." the authors aim to critically analyze the acceptability of COVID-19 vaccination among dental students and dental practitioners. Although, the authors have addressed all the previous comments raised by different reviewers. There are a few concerns on which authors may work to improve the manuscript.

Comments:

1) In the study selection section, the authors mention one of the exclusion criteria as " poor data reported on acceptance or hesitancy level." Please report how authors have differentiated between poor data and sound data. Also, what parameters are considered to determine this selection?

2) The English of the manuscript may be improved. There are instances in the manuscript where the construction of sentences may be better (Minor).

3) The studies considered for the study are from different countries. The number of COVID-19 cases (danger) and the availability of the vaccine to citizens are different in each country. Please discuss these factors in the discussion.

Reviewer #2: (No Response)

Reviewer #3: Abstract

- Maximum word count is 300. Again, please shorten, and note that this is due to the various databases, and must be revised.

All other aspects have been discussed/answered. Draft is ready for external review.

7. PLOS authors have the option to publish the peer review history of their article (what does this mean?). If published, this will include your full peer review and any attached files.

Reviewer #1: **Yes: **Aditya Kumar Sharma

Reviewer #2: **Yes: **Dr. Deepak Kumar Padhi

Reviewer #3: No

---

## [Author Response · Author response to Decision Letter 2]

5 Apr 2022

1. Please review your reference list to ensure that it is complete and correct. If you have cited papers that have been retracted, please include the rationale for doing so in the manuscript text or remove these references and replace them with relevant current references. Any changes to the reference list should be mentioned in the rebuttal letter that accompanies your revised manuscript. If you need to cite a retracted article, indicate the article’s retracted status in the References list and also include a citation and full reference for the retraction notice 

Reply: 

The reference list was amended accordingly. 

The second reference (No.2) has been changed.

Reviewer 1

1. In the study selection section, the authors mention one of the exclusion criteria as " poor data reported on acceptance or hesitancy level." Please report how authors have differentiated between poor data and sound data. Also, what parameters are considered to determine this selection? 

Reply:

The authors have amended the following sentence for better understanding:

“Mean and standard deviation on the acceptance or hesitancy level are not reported”

2. The English of the manuscript may be improved. There are instances in the manuscript where the construction of sentences may be better (Minor). 

Reply:

The manuscript has been proofread by a native English speaker.

3. The studies considered for the study are from different countries. The number of COVID-19 cases (danger) and the availability of the vaccine to citizens are different in each country. Please discuss these factors in the discussion. 

Reply:

The authors appreciate the suggestion raised by the reviewer and have included some points in the discussion:

“Furthermore, despite growing evidence of the safety and effectiveness of presently used vaccines …… continue to rise as vaccine availability increases [37].”

Reviewer 3

1. Maximum word count is 300. Again, please shorten, and note that this is due to the various databases and must be revised. 

Reply: 

The authors have revised the abstract to 300 words.

---

## [Decision Letter · Decision Letter 3]

7 Apr 2022

COVID-19 Vaccination Acceptance Among Dental Students and Dental Practitioners: A Systematic Review and Meta-Analysis

PONE-D-22-00027R3

Dear Dr. Lin,

We’re pleased to inform you that your manuscript has been judged scientifically suitable for publication and will be formally accepted for publication once it meets all outstanding technical requirements. Congratulations, and best wishes!

Prof. Dr. med. dent. Dr. h. c. Andrej M Kielbassa

Kind regards,

Prof. Dr. med. dent. Dr. h. c. Andrej M Kielbassa

Academic Editor

PLOS ONE

Additional Editor Comments (optional):

Reviewers' comments:

Reviewer's Responses to Questions

**Comments to the Author**

1. If the authors have adequately addressed your comments raised in a previous round of review and you feel that this manuscript is now acceptable for publication, you may indicate that here to bypass the “Comments to the Author” section, enter your conflict of interest statement in the “Confidential to Editor” section, and submit your "Accept" recommendation.

Reviewer #1: All comments have been addressed

Reviewer #3: All comments have been addressed

2. Is the manuscript technically sound, and do the data support the conclusions?

Reviewer #1: Yes

Reviewer #3: Yes

3. Has the statistical analysis been performed appropriately and rigorously? 

Reviewer #1: Yes

Reviewer #3: Yes

4. Have the authors made all data underlying the findings in their manuscript fully available?

Reviewer #1: Yes

Reviewer #3: Yes

5. Is the manuscript presented in an intelligible fashion and written in standard English?

Reviewer #1: Yes

Reviewer #3: Yes

6. Review Comments to the Author

Reviewer #1: The author has addressed all my comments. Therefore, there is no basis on which manuscript can be rejected for publication.

I congratulate the authors for the work.

Reviewer #3: With the help of the reviewers, this revised and re-submitted paper has been considerably improved, and is ready to proceed.

7. PLOS authors have the option to publish the peer review history of their article (what does this mean?). If published, this will include your full peer review and any attached files.

Reviewer #1: **Yes: **Aditya Kumar Sharma

Reviewer #3: No

---

## [Editor Report · Acceptance letter]

11 Apr 2022

PONE-D-22-00027R3 

COVID-19 Vaccination Acceptance Among Dental Students and Dental Practitioners: A Systematic Review and Meta-Analysis 

Dear Dr. Lin:

I'm pleased to inform you that your manuscript has been deemed suitable for publication in PLOS ONE. Congratulations! Your manuscript is now with our production department. 

Kind regards, 

on behalf of

Prof. Dr. med. dent. Dr. h. c. Andrej M Kielbassa 

Academic Editor

PLOS ONE